# Molecular Epidemiology of the Main Druggable Genetic Alterations in Non-Small Cell Lung Cancer

**DOI:** 10.3390/ijms22020612

**Published:** 2021-01-09

**Authors:** Sara S. Fois, Panagiotis Paliogiannis, Angelo Zinellu, Alessandro G. Fois, Antonio Cossu, Giuseppe Palmieri

**Affiliations:** 1Department of Medical, Surgical and Experimental Sciences, University of Sassari, Viale San Pietro 43, 07100 Sassari, Italy; sara.solveig.fois@gmail.com (S.S.F.); agfois@uniss.it (A.G.F.); cossu@uniss.it (A.C.); 2Department of Biomedical Sciences, University of Sassari, Viale San Pietro 43b, 07100 Sassari, Italy; azinellu@uniss.it; 3Unit of Cancer Genetics, Institute of Genetic and Biomedical Research (IRGB), National Research Council (CNR), Traversa La Crucca 3, 07100 Sassari, Italy; gpalmieri@yahoo.com

**Keywords:** lung, NSCLC, *EGFR*, *ALK*, *ROS1*, targeted therapy

## Abstract

Lung cancer is the leading cause of death for malignancy worldwide. Its molecular profiling has enriched our understanding of cancer initiation and progression and has become fundamental to provide guidance on treatment with targeted therapies. Testing the presence of driver mutations in specific genes in lung tumors has thus radically changed the clinical management and outcomes of the disease. Numerous studies performed with traditional sequencing methods have investigated the occurrence of such mutations in lung cancer, and new insights regarding their frequency and clinical significance are continuously provided with the use of last generation sequencing technologies. In this review, we discuss the molecular epidemiology of the main druggable genetic alterations in non-small cell lung cancer, namely *EGFR*, *KRAS*, *BRAF*, *MET*, and *HER2* mutations or amplification, as well as *ALK* and *ROS1* fusions. Furthermore, we investigated the predictive impact of these alterations on the outcomes of modern targeted therapies, their global prognostic significance, and their mutual interaction in cases of co-occurrence.

## 1. Introduction

Lung cancer (LC) is one of the most incident malignancies worldwide. According with the Global Cancer Observatory (GCO), more than 2,200,000 new cases and approximately 1,800,000 deaths were estimated in the world in 2020 [1]. Higher age-adjusted incidence rates were estimated in North America and Europe (32.6 and 29 cases, respectively, per 100,000 inhabitants), while higher mortality rates were observed in Europe (22.6 per 100.000); mortality rates were similar in North America and China [1]. The closeness between incidence and mortality rates shows that LC remains highly lethal, despite recent improvements in the prevention, screening, diagnosis, and clinical management of the disease.

Cigarette smoking is considered to be the main determinant of the distribution of LC cases among genders and populations. In the past few decades, LC has been reported to be more common among males who were most commonly smokers; nevertheless, recent reports describe a decline in LC cases in men and a consistent emerging increase in women, especially in young ages [2]. A report from the United States reported higher LC incidence in young women than young men [3], and this trend does not seem to exclusively depend on the changing patterns in cigarette smoking [2]. Other known risk factors for lung cancer include exposure to secondhand smoke, mineral, metal dusts, and radon, as well as asbestos, which is the main risk factor for the development of malignant mesothelioma [4]. With the introduction of screening guidelines and decrease in tobacco use in the United States, the mortality rate for lung cancer has recently decreased by 48% in males and 23% in females [5], while a significant increase in the LC five-year survival rates by an average of 1.3% a year has been observed in China over the past 12 years [6].

From a pathological perspective, LC has been classically divided in two major classes: small cell lung cancer (SCLC) and non-small cell lung cancer (NSCLC). The latter comprises the most common histological subtypes, such as adenocarcinoma, squamous cell carcinoma, and large cell carcinoma. Squamous cell carcinoma was the most frequent NSCLC histotype until the 1980s, when it was superseded by adenocarcinoma [7], probably due to changes in the characteristics of cigarettes (increased puff volume and nitrate levels), and the aforementioned incidence increase in women who tend to be mostly affected by adenocarcinomas [4]. NSCLC accounts for approximately 85% of the LC cases currently observed worldwide [8].

Treatment strategies and prognoses of NSCLC largely depend on the stage of the disease at diagnosis. Surgery is highly curative in early stages but totally ineffective in patients with metastasis; unfortunately, the latter comprise the greatest part of the cases at the time of diagnosis. In this subset of patients, the advent of targeted agents represents the most important innovation over the last years. The discovery of activating mutations of the epidermal growth factor receptor (*EGFR*) in patients with lung adenocarcinoma led to the development of a new family of biological agents, called tyrosine kinase inhibitors (TKIs) [9]. These medications have revolutionized the clinical management of patients harboring *EGFR* mutations, whose survival nearly doubled compared to standard chemotherapy [8]. The increased frequency of resistance to TKIs redimensioned initial enthusiasm and led to the progressive discovery of novel agents targeting different molecular alterations and pathways. Currently, it is estimated that up to 69% of advanced NSCLC patients have druggable mutations in numerous genes, including *EGFR*, anaplastic lymphoma kinase (*ALK*), c-ros oncogene 1 (*ROS1*), Kirsten rat sarcoma virus (*KRAS*), V-raf murine sarcoma oncogene homolog B1 (*BRAF*), *MET*, human epidermal growth factor receptor (*HER2*), and other genes [10]. Nevertheless, to date, only drugs targeting *EGFR* and *BRAF* mutations, and *ALK* or *ROS1* rearrangements have been approved for clinical use. In particular, the third generation anti-*EGFR* osimertinib is currently recommended for patients with *EGFR*-mutated lung adenocarcinoma, alectinib, or brigatinib for those harboring *ALK* rearrangements, and crizotinib for those with *ROS1* rearrangement. In addition, patients with tumors harboring *BRAF* V600E mutations can be treated with a combination of dabrafenib (anti-*BRAF*) plus trametinib (anti-*MEK*). The list will be surely expanded in the near future because intensive research is ongoing and better-designed medications, like larotrectinib for *NRTK*-altered cases, selpercatinib for *RET*-mutated cases, and others in *KRAS*-mutated cases, showed encouraging results in recent clinical trials [11,12,13]. Recent reports evidenced that the epidemiological patterns of these and other targetable genetic alterations vary considerably throughout the world and among different geographical areas, populations, genders, and individuals with different characteristics or habits.

The aim of this review was to summarize the global epidemiology of the main druggable molecular alterations in NSCLC, as well as their correlation with the demographic, anthropometric, pathological, and clinical data of the affected individuals.

## 2. Molecular Epidemiology of the Main Druggable Genes in LC

### 2.1. EGFR

*EGFR*, also known as *RBB*, *ERBB1*, and *HER1*, is a gene located in the short arm of chromosome 7. It encodes a transmembrane protein with a large extracellular component (four domains and ~620 amino acids) that primarily serves as ligand-binding sites and that is anchored by a short helical transmembrane domain to the intracellular tyrosine kinase domain (TKD) [14].

This protein is a member of a family of four structurally related tyrosine kinase receptors (RTKs) that includes human epidermal growth factor receptor 2 (ERBBB2/HER2), ERBB3/HER3, and ERBB4/HER4. *EGFR*, upon activation by specific ligands, homo- or hetero-dimerizes at the cell membrane and triggers the receptor’s intrinsic tyrosine kinase activity with the autophosphorylation of the tyrosine residues in the C-terminal domain of the protein itself [9]. This results in the downstream activation of other signaling proteins that are associated with several transduction cascades. These cascades include the mitogen-activated protein kinase/extracellular signal-related kinase (MAPK/ERK) pathway, the PI3K-AKT-mTOR pathway, the JNK pathway, and others implicated in cell migration, adhesion, and proliferation pathways (Figure 1) [15].

Abnormalities within these pathways have been observed in NSCLC and are seen in tumors carrying mutations of the *EGFR* gene. In these neoplasms, *EGFR* plays a pivotal role in cellular proliferation, the inhibition of apoptosis, angiogenesis, and metastatic progression and chemoresistance. In NSCLC, *EGFR* mutations are limited to the first four exons (exons 18–21) of the tyrosine kinase domain, which encode the N-lobe and the 5′ portion of the C-lobe of *EGFR*; they consist of three different types of mutations (deletions, insertions, and missense point mutations), and they all cluster around the ATP binding pocket of the TKD of *EGFR* [16,17]. These mutations destabilize the autoinhibited conformation of the receptor, and the protein is then constitutionally activated to produce its pro-oncogenic effects [18]. On the other hand, these mutations are associated with hypersensitivity to TKIs, as we mentioned above, and are thus termed “sensitizing” *EGFR* mutations.

The prevalence of *EGFR* mutation varies with histotype, ethnicity, and other demographic or pathological factors. Several studies have shown that sensitizing *EGFR* mutations are almost exclusively observed in NSCLC patients with adenocarcinoma, rather than in those with other histologies; this observation is even more striking in East Asian populations, where *EGFR* mutations are present in up to 78% of adenocarcinomas, as opposed to only 10–16% of adenocarcinomas in other ethnicities [19,20,21]. The reasons for such a discrepancy in mutational rates are unclear, with some studies suggesting alternative mechanisms that may be required for the development of lung cancer in Asians, who seem to have an inherent non-environmental susceptibility to development of *EGFR* mutations [22,23]. Frequencies in Middle East and African populations are slightly higher than those observed in Western populations but still lower than in Asian populations. A slightly lower prevalence has been observed in the Oceanic ethnicities and other insular Mediterranean populations (12%) [19,24,25].

About 90% of NSCLC-specific sensitizing *EGFR* mutations are either in-frame microdeletions around the Leu–Arg–Glu–Ala (LREA) residues of exon 19 (known as Del19) or missense leucine-to-arginine substitution at codon 858 (L858R) in exon 21 [26,27]. Del19 alone, of which at least 30 variants have been identified, accounts for 45–60% of all mutations, while L858R accounts for 35% to 45% of all mutations. Globally, these two types of mutations are termed “classic” or “common” *EGFR* mutations, and they are thought to produce similar configurational changes within the ATP binding cleft of the protein [16]. The remaining *EGFR*-mutant patients carry “uncommon” mutations. These comprise a heterogeneous group and have been found in 7–23% of patients with *EGFR* mutant NSCLC [17], even though there are discrepancies across studies and their true individual frequency remains to be better determined. They can be isolated or occur together with an independent *EGFR* mutation (“complex” or “compound” mutations). The most frequently identified uncommon mutations are G719X, S768I, and L861Q in exons 18, 20, and 21, respectively. Larger studies performed with modern sequencing techniques have detected rarer mutations, such as A750P, T790M, L62R, S752F, Del18, and, more recently, even exon 18−25 kinase domain duplications and *EGFR* fusion events. Some studies have reported high frequencies (23–43% of all uncommon mutations) of Ins20 mutations [17].

Regarding demographic factors, approximately all published studies have reported a higher frequency of *EGFR* mutations in women versus males, with figures of up to 69.7%. In other words, up to 42% of females versus only 14% of males with NSCLC are expected to harbor an *EGFR* TK domain mutation [21,28]. Regarding patient age at diagnosis, multiple targetable genomic alterations (including *EGFR*) have been found to be more prevalent in patients diagnosed at a younger age, although some controversial findings exist. Smoking status is another independent factor associated with the *EGFR* mutation incidence that is higher in never smokers (up to 66.6% of the cases) [28,29]. This was confirmed in a recent meta-analysis that included 167 epidemiologic studies with over 63,000 LC cases to calculate summary odds ratios for specific mutations (including *EGFR*) in never and ever smokers [30]. The meta-analysis also showed that as the smoking history increased, there were decreased odds for exhibiting the *EGFR* mutation, particularly for cases >30 pack-years. This is particularly true for Caucasian non-smokers, where the mutation prevalence seems to be relatively greater than in Asian non-smokers [21,25]. Table 1 summarizes the main epidemiological data available regarding *EGFR* and the other driver genes discussed in the present review.

From a clinical point of view, tumor stage at diagnosis does not generally seem to correlate with the presence of *EGFR* mutations [28,31,32], but such mutations have been linked to the presence of multiple lung lesions and air bronchograms on imaging [23]. Older data showed that ever-smokers with *EGFR*-mutated lung adenocarcinoma tend to be diagnosed with more advanced pathologic stage disease [16].

As several randomized trials have extensively demonstrated, the presence of a classic sensitizing *EGFR* mutation is the most powerful predictive biomarker of response to monotherapy with first-generation TKIs and, thus, the most important prognostic factor in cases of advanced lung adenocarcinoma [25,33,34,35]. In this subset of cases, poorer outcomes have been associated with male sex, the presence of the L858R mutation, and the diagnosis of the former bronchioalveolar adenocarcinoma [29]. Exon 18 aberrations also appear to confer poorer prognosis in relation with other *EGFR* mutations [36]. Despite the initial effectiveness of TKIs, all *EGFR*-mutated tumors eventually progress due to the occurrence of secondary mutations in the *EGFR* TKD or in genes involved in alternative molecular pathways. In about 50% of cases, it is the so-called gatekeeper T790M, also known as Thr790Met, that causes drug resistance by increasing the affinity of the domain for ATP [37,38]. T790M can also be present de novo, as a co-occurring mutation, in a small proportion of *EGFR*-positive patients, explaining primary resistance to first generation TKIs. Second-generation irreversible TKIs target the acquired T790M mutation. A few other unclassical *EGFR* mutations are predictive of resistance to treatment with TKIs, especially exon 20 insertions [39,40,41].

### 2.2. ALK

The *ALK* gene consists of 30 exons mapping to the long arm of chromosome 2 (2p23.2–p23.1). It encodes a transmembrane protein of 1620 amino acids that belongs to the insulin receptor superfamily of RTKs, and it contains an extracellular domain, a transmembrane domain, and an intracellular domain [42]. *ALK* plays essential roles in the development of the brain, but it is also expressed in scattered adult cells such as neurons, glial cells, the testes, the pituitary gland, the hypothalamus, and endothelial cells [43]. Even though *ALK* is considered an orphan receptor, pleiotrophin and midkine are known ligands. Unlike for other receptors, the mechanisms for the activation of *ALK* are not fully understood. Triggering by the binding of the ligands to the extracellular domain seems the most likely, as well is the subsequent stimulation of classical signaling cascades such as MAPK, PI3K/mTOR, JAK-STAT, and SHH. In pathologic conditions, the downstream activity of these pathways leads to increased cell proliferation and metabolism, migration, survival, and apoptosis avoidance [43].

*ALK* aberrations in NSCLC were first identified in 2007. Most alterations are chromosomal rearrangements that result in the formation of fusion genes through small inversions within chromosome 2 [42]. The break points for the translocations of *ALK* genes are typically located at exons 19–20 or exons 20–21. Fusion genes comprising echinoderm microtubule-associated protein like 4 (EML4) and the *ALK* genes were the first identified, as well as the most frequent in NSCLC, accounting for about 80% of all fusion events [44]. Generally, the resulting chimera proteins contain the complete kinase domain and have a potent oncogenic effect due to the accelerated tyrosine kinase activity mediated by the fusion partner [45]. Several EML4-*ALK* fusion variants have been identified in NSCLC, differing in the breaking point between the two genes [46]. Variants 1, 2, and 3a/3b are the most common, accounting for more than 80% of all EML4-*ALK* variants, regardless of clinical characteristics such as gender, smoking status, histology, and stage [46,47]. Variant 1 alone accounts for about 60% of all EML4-*ALK* [47]. It consists of a simple fusion between exon 13 of EML4 and exon 20 of the *ALK* gene [48]. Rarer *ALK* fusion partners in NSCLC are *TFG*, *KIF5B*, *KLC1*, *STRN*, *TPR*, *HIP1*, *GCC2*, *DCTN1*, *SQSTM1*, *LMO7*, *BIRC6*, *PHACTR1*, and *PTPN3* [44,45].

*ALK* aberrations act as oncogenic drivers in 1–10% of NSCLC cases according to recent epidemiological studies [25,43,44,49,50], but the incidence rates of such aberration are significantly variable among different populations. One study reported a higher prevalence of *ALK*-positive disease in Asian (22.0%), Pacific (10.8%), and Māori (6.9%) ethnicity patients in comparison with New Zealand Europeans (4.4%) [49]. By contrast, other studies have reported similar frequencies among different ethnicities [48,50]. A relatively high prevalence of *ALK*-positive cases (13%) was found in a selected population of Asian females with never/light smoking history and adenocarcinoma histology [51].

*ALK* translocations are more prevalent among individuals in their fourth or fifth decade of life, considerably younger than both patients with NSCLC (median age of 70 years) and patients with tumors harboring *EGFR* mutations (60–65 years) [49,52]. Colombino et al. found a significantly higher incidence of *ALK* rearrangements in patients with lung adenocarcinoma at less than 50 years of age in comparison to older individuals [25]. This is in accordance with epidemiological data regarding other cancers known to harbor high rates of *ALK* rearrangements, such as anaplastic large cell lymphomas and neuroblastomas, and involve commonly children and young adults [51]. Nevertheless, *ALK* translocations have been found in both patients aged younger than 40 and older than 70 in approximately all the studies performed. Never or light smokers are also more likely to carry an *ALK* rearrangement; in published studies, up to 70–80% of patients are never smokers, while no clear gender predilection has emerged to date [42,44,46,49,50,51].

Regarding histology, *ALK* rearrangements are typically detected in lung adenocarcinomas with solid growth predominance. In particular, data from cytomorphologic analysis positively associate extracellular mucin, cribriform structure, signet ring cells, and hepatoid cytology with the presence of *ALK* rearrangements [49]. Acinar growth and bronchioalveolar patterns are rarely documented in *ALK*-rearranged tumors. Despite their large predominance in adenocarcinomas, *ALK* rearrangements are also sporadically reported in lung squamous cell carcinoma (SCC) [46]. In addition, considering the difficulties in establishing the exact histotype in small biopsies, it is currently recommended to perform *ALK* testing in tumors of a potentially mixed (adenosquamous) histology, except in those with adenocarcinoma.

It has been evidenced that *ALK*-positive NSCLC has a highly aggressive clinical behavior and the tendency to be found at a higher clinical stage at diagnosis in comparison with wild-type patients [42,48,51,52]. Numerous trials have showed that the presence of *ALK* rearrangements is a dominant predictive factor of response to treatments with TKIs [49]. However, heterogeneous responses have been reported, and almost all *ALK*-rearranged tumors eventually experience resistance to crizotinib [44]. The sequencing of resistant tumor DNA allowed for the identification of resistant *ALK* point mutations in 20–40% of the cases [53]. The L1196M gatekeeper mutation (analogous with the *EGFR* T790M mutation) and the G1269A mutation are the most frequent in patients resistant to crizotinib, while the highly resistant G1202R mutation accounts for less than 10% of cases [43]. Other concurrent somatic mutations have recently been identified (T1151R, R1192P, A1280V, and L1535Q) to confer strong resistance to *ALK*-TKIs [44]. In addition, EML4-*ALK* patients show lower response rates to platinum-based chemotherapy when compared to those with *EGFR* mutations [54]. The short translocation forms (variant 3 and others) seem to be characterized by a more aggressive oncological behavior and lower sensitivity to *ALK*-TKIs than cases with long forms (variant 1 and others). Second generation *ALK* inhibitors that may overcome some of these resistances have been recently introduced in clinical practice.

### 2.3. ROS1

*ROS1* is a gene located in chromosome 6q22 [55]. It encodes a sizable 2347-amino-acid-long transmembrane receptor with an extracellular N-terminal domain, a hydrophobic transmembrane region, an intracellular C-terminal tyrosine kinase domain, and a carboxy terminal tail [56]. *ROS1* shares many structural and functional similarities with the *ALK* protein, but it has a unique and large extracellular domain containing six repeat motifs with high homology to the extracellular matrix, as well as fibronectin type III-like sequence repeats, that resembles a cell adhesion molecule [57]. *ROS1* functions as an orphan RTK receptor, with an unknown ligand. As a consequence, little is known about its function; it seems that it is able to directly couple extracellular adhesion-mediated events to intracellular signaling mediated by tyrosine phosphorylation [58]. Wild-type *ROS1* activation has been implicated in the mesenchymal-to-epithelial transition and in cellular differentiation during the development of kidney, lung, small intestine, heart, and lung cancer [56]. The dysregulation of the *ROS1* kinase activity via the acquisition of certain gene fusions leads to the activation of the downstream cascades of several oncogenic pathways such as PI3K-Akt, mTOR, RAS-MAPK/ERK, VAV3, SHP-1 and SHP-2, which variably impact cell differentiation, proliferation, growth, and survival [56,58,59,60].

*ROS1* gene fusions were first identified in the human glioblastoma cell line U-118 MG*ROS1* and, subsequently, in lung cancer in 2007 [61]. Their overall frequency in NSCLC ranges from 0.9% to 2.6%, with similar rates across Asia, Europe, and North America and a worldwide prevalence estimated as 1.9% [57,59]. Thus far, several fusion patterns have been described. CD74-*ROS1* was the first *ROS1* rearrangement identified in NSCLC and is by far the most frequently reported, consisting of 32% of all *ROS1* fusions [57,61]. Other common translocations include SLC34A2-*ROS1* (~17%), tropomyosin 3 (TPM3-*ROS1*; ~15%), syndecan 4 (SDC4-*ROS1*; ~11%), ezrin (EZR-*ROS1*; ~6%), and FIG-*ROS1* (~3%). Further rearrangements such as leucine-rich repeats and immunoglobulin-like domains 3 (LRIG3-*ROS1*), endoplasmic reticulum protein retention receptor 2 (KDELR2-*ROS1*), coiled-coil domain containing 6 (CCDC6-*ROS1*), Golgi-associated PDZ and coiled-coil motif containing gene (GOPC-*ROS1*), moesin gene (MSN-*ROS1*), and clathrin heavy chain gene (CLTC-*ROS1*) occur in less than 1% of the cases [59,62].

*ROS1* epidemiological patterns are very similar to those of *ALK* translocations, as it is more common in young individuals and never smokers, as confirmed in a recent meta-analysis [63]. Nevertheless, *ROS1* rearrangements seem to involve more commonly female patients, resembling, in this case, the *EGFR* mutations. In addition, *ROS1* fusions are more prevalent in adenocarcinomas than in other NSCLC histologies [57,62,63,64]. Interestingly, the cytomorphologic features (extracellular mucin, cribriform structure, signet ring cells, and hepatoid cytology) of adenocarcinomas harboring *ROS1* translocations are very similar to those described in *ALK*-rearranged cases, suggesting that fusion genes in lung adenocarcinomas have specific identifiable morphological features [47].

*ROS1* rearrangements are significantly more frequent in advanced NSCLC clinical stages [63], but they were associated with lower rates of extrathoracic disease, including brain metastases, at initial metastatic diagnosis [65]. Patients with *ROS1*-rearranged disease are expected to respond to TKIs as much as *ALK*-rearranged tumors, and crizotinib was granted full approval in 2016 [66,67]. As for other TKIs, however, patients ultimately develop drug resistance. Discrepancy in sensitivity to *ALK*/ROS TKIs among different *ROS1* fusion patterns has been described, but the underlying mechanisms remain to be elucidated [57].

### 2.4. KRAS

The *KRAS* gene is located on the short arm of chromosome 12 (12p12.1) and encodes a guanosine triphosphatase (GTPase) that links cell surface receptors to signaling pathways such as rapidly accelerated fibrosarcoma (RAF)-MEK-ERK, PI3K-AKT-mTOR, and RALGDS-RA [68]. *KRAS* is a proto-oncogene and acts as a molecular on/off switch regulator to cell proliferation, maturation, and differentiation. At the cell level, binding of an appropriate ligand (i.e., *EGFR*, *ALK,* or MET) with the *KRAS* protein leads to its dimerization, phosphorylation, and, thus, activation. RAS phosphorylation is determined by a balance of pro-phosphorylating guanine nucleotide exchange factors (GEFs) and negatively regulating GTPase activating proteins (GAPs). Activated RAS can activate downstream effectors belonging to the previously mentioned pathways. *KRAS* is the most frequent oncogene in cancer [68]. *KRAS* mutations have been found in numerous malignancies, including gastric, biliary, skin, and gynecological cancer, and they are the most common genetic alterations in pancreatic, colorectal, and lung cancer [25,69,70,71,72,73]. In NSCLC, *KRAS* mutations occur in about one third of the cases, thus representing the second most common genetic alteration after *p53* mutations [74,75]. The vast majority of activating *KRAS* mutations in NSCLC are found in codons 12 and 13 (rarely in codon 61) and change the amino acid glycine in the *KRAS* protein [76]. They include G12C, G12V, and G12D. These mutations result in the loss of intrinsic GTPase activity, the persistence of the GTP-bound state, and, consequently, the deregulation of cell proliferation signals [77].

In NSCLC, as opposed to *EGFR* mutations, *KRAS* mutations are more common in Western than in Asian or Australian populations (23–33% vs. 2–15%, respectively, of cases) and in long-term tobacco smokers than in never-smokers (20–44% vs. 6–10%, respectively) [25,78,79,80,81]. *KRAS* transition mutations (G → A) are more common in patients with adenocarcinoma and no smoking history, whereas transversion mutations (G → T or G → C) are associated with tobacco exposure. This observation suggests that *KRAS*-mutated NSCLC in non-smokers may not be caused by second-hand tobacco exposure [82]. Unlike *EGFR* aberrations, *KRAS* mutations do not seem to have any gender or age predilection, even though young women have been reported to display a higher susceptibility to the transversion mutation G12C. This mutation is the single most frequent mutation among smokers, suggesting an increased susceptibility to tobacco carcinogenesis in women compared to men [83]. In contrast, Colombino et al. found a significantly higher incidence of *KRAS* mutations in males in a recent study performed in Italy [25].

The prognostic impact of *KRAS* mutations in NSCLC is not clear. Several individual studies and a metanalysis of 28 studies showed that tumors harboring somatic *KRAS* mutations have a worse prognosis and a reduced or absent response to *EGFR* TKIs [74,79,84,85,86,87]. Nevertheless, these results were not confirmed in other studies; for example, in the JBR.10 randomized controlled trial, *KRAS* mutations were not predictive of survival (HR for OS: 1.23, 95% CI: 0.76–1.97) [88]. In addition, *KRAS* has not been yet targeted directly in lung cancer, and attempts to inactivate either upstream and downstream proteins involved in RAS signaling have been met with limited success to date.

### 2.5. BRAF

*BRAF* is a gene mapping on chromosome 7q34 encoding a protein that is a member of the RAF kinase family, which comprises three serine-threonine kinases: ARAF, BRAF, and CRAF. The *BRAF* protein contains 766 amino acids and is arranged into three highly conserved regions called CR1, CR2, and CR3. CR1 is located near the N-terminus and comprises both a cysteine-rich domain and a Ras-binding domain, while CR2 links the CR1 and CR3 domains and contains a binding site for the 14-3-3 inhibitory protein. Finally, CR3, which resides close to the C-terminus, contains a kinase domain that phosphorylates and activates downstream targets [89]. In addition, CR1 interacts with CR2 to disable the activation of the protein, which requires the binding of Ras-GTPases at the cellular membrane. Once activated, the *BRAF* protein forms homodimers with itself and heterodimers with other RAF kinases, and it plays important roles in regulating the MAPK/ERK signaling pathway, ultimately inducing cell growth, mobility, and survival [90].

The oncogenic potential of *BRAF* mutations was first described in 2002 by Davies et al. in melanoma and NSCLC cell lines [91]. Subsequent studies showed that mutations in *BRAF* result in the constitutive activation of the MAPK/ERK signaling pathway, leading to uncontrolled cellular proliferation and cell survival. They have been detected in various tumors, including NSCLC. *BRAF* mutations are involved in approximately half the cases of malignant cutaneous melanomas, as well as in sinonasal and other melanomas to a lesser extent, and represent a paradigm of successful molecular targeting therapies [92,93]. Other cancers in which *BRAF* is recurrently mutated include papillary thyroid carcinoma, hairy cell leukemia, and colorectal cancer [94,95,96]. In NSCLC, *BRAF* mutations occur in 2–4% of cases, with similar figures in Asian and Caucasian populations [96]. *BRAF* mutations are typically distributed throughout exons 11 and 15, with most of them occurring at exon 15, which encodes the catalytic domain of the kinase. They have been classified into three functional classes based on signaling mechanism, kinase activity, and sensitivity to inhibitors: RAS-independent kinase-activating V600 monomers (class I), RAS-independent kinase-activating dimers (class II), and RAS-dependent kinase-inactivating heterodimers (class III) [97,98]. Class I V600E mutations are the most predominant, as they account for 20–30% of all *BRAF* mutations. They result in a strong activation of the MAPK/ERK pathway, regardless of RAS, whose activation is suppressed through a negative feedback loop triggered by ERK activation, involvement. Class II (L597Q/R, G464V/A, G469A/V/R/S, K601E/N/T, E451Q, A712T, and fusions) mutations hold an intermediate kinase activity and RAS independence; they account for about 20% of all *BRAF* mutations. Class III (G469E, G466V/E/A, N581S/I, D594G/N, and G596R) mutations rely on RAS activation to overcome negative feedback from ERK [97,98].

Globally, *BRAF* mutations have a strong male (61%) and ever-smoker predominance (81%), with variable differences among different mutational classes [99]. Class I mutations, for instance, are more likely to occur in females, whereas males are more likely to harbor non-V600 mutations [97,98,100]. In addition, several studies have shown that class I mutations are more frequent among never-smokers in comparison with class II or III [98,101]. No correlation with age at diagnosis has been reported so far [100]. *BRAF* mutations are almost exclusively detected in adenocarcinomas, with only anecdotal reports in SCC [98,100,101].

Regarding prognosis, class I mutations appear to have a slightly better prognosis than their class II and III counterparts, which have been associated with a more aggressive behavior and less favorable clinical course, as well as earlier disease progression after first-line chemotherapy [98,101]. However, these findings were not confirmed in all studies, and whether *BRAF* mutational status actually affects oncological outcomes remains to be clarified. What is certain is that the presence of the *BRAF* V600 mutation in NSCLC is a predictive marker of response to specific RAF inhibitors, such as vemurafenib or dabrafenib, in monotherapy or in combination with an MEK inhibitor [98,99,100]. In addition, it has been observed that MAPK alterations, including *BRAF* and *SHP2* mutations, are more frequent in patients who respond to PD-1 checkpoint inhibitors like nivolumab and pembrolizumab [102]. This may depend on the fact that these mutations are more frequently observed in smokers with a higher mutational load and thus a greater benefit with immunotherapy. Non-V600 mutations are currently not eligible for targeted therapies; however, interdependence from RAS suggests that class III *BRAF* mutant tumors could be sensitive to RTK inhibitors [97].

### 2.6. MET

The *MET* gene, also known as hepatocyte growth factor receptor (HGFR) resides in the long arm of the chromosome 7 (7q21–q31) and encodes an RTK for the hepatocyte growth factor (HGF). After posttranslational cleavage and glycosylation, a 50-kDa alpha chain and a 140-kDa beta chain are produced. The alpha chain is linked to the extracellular portion of the beta chain, which contains the semaphoring domain, a plexin-semaphorin-integrin (PSI) domain, four immunoglobulin-plexin-transcription (IPT) repeats, a transmembrane domain, a juxta membrane domain, a tyrosine kinase domain, and the C-terminus. Similarly to other previously described proteins, MET also dimerizes upon binding to HGF, resulting in the phosphorylation of the docking sites for adaptor proteins that activate the downstream signaling to pathways such as MAPK/ERK and PI3K/AKT [103]. The intracellular juxta membrane domain of the protein is encoded by exon 14 and contains critical regulatory elements, including the direct binding site for the ubiquitin ligase that promotes MET protein degradation [104]. Mutations in the *MET* gene can cause exon 14 skipping, and the resulting mutant receptor demonstrates enriched signaling and oncogenic potential, due to loss of a portion of its juxta-membrane domain.

*MET* aberrations have been well-documented in multiple oncogenic processes. In NSCLC, the three main mechanisms of *MET* dysregulation are protein overexpression and gene amplification or mutation. Gene rearrangements have been reported anecdotally [103]. The reported prevalence of driver mutations in the splice site of *MET* that result in exon 14 skipping in NSCLC ranges from 1% to 10%, with lower figures among Asian compared to Western populations [105,106]. A consistently greater incidence of *MET* amplifications has been extensively documented as a mechanism of acquired resistance in 5–22% *EGFR*-mutated NSCLC upon therapeutic pressure with *EGFR*-TKIs [105]. Nevertheless, the incidence of *MET* amplification is significantly lower, ranging from 2% to 5% in naïve NSCLC patients, as reported by Colombino et al. in a study performed on 1440 patients with adenocarcinoma [75,103,107,108]. This confirms that pharmacologic pressure acts as a triggering mechanism of *MET* amplifications. *MET* amplification stimulates in vitro cell proliferation and migration, promoting the progression of the disease and metastases formation in *EGFR*-mutated NCSLC tumors [109]. In addition, *MET* gene amplification causes first generation *EGFR*-TKI resistance by activating the *EGFR*-independent phosphorylation of ERBB3 and the downstream activation of the PI3K/AKT pathway, thus providing a bypass mechanism.

*MET*-mutated NSCLC has been correlated with an older age at diagnosis compared with *EGFR*, *KRAS*, and *BRAF*-mutant lung cancers, with a median age of 72.5 years; furthermore, two thirds of patients harboring *MET* exon 14 mutations have been reported to be current or former smokers, but in a recent Italian study, no differences in *MET* amplifications in relation with the smoking status were found in naïve patients with lung adenocarcinoma [25]. *MET* amplification predominately occurs in adenocarcinomas and also seems to have correlated with an earlier stage of the disease in some studies, but several authors have reported conflicting results [105,110]. In the study of Go et al. *MET* amplifications occurred in 3.9% of the 180 enrolled NSCLC patients and were more common in SCC than in those with adenocarcinoma [111]. NSCLC-harboring *MET* alterations are sensitive to treatment with inhibitors such as crizotinib and cabozantinib, and the therapeutic effect of these drugs is proportionally correlated with the levels of amplification; indeed, patients with higher levels of *MET* amplification are also expected to have longer progression-free survival (PFS) [105,110]. Nevertheless, in naïve patients, a higher *MET* gene copy number has been associated with worst outcomes [107].

### 2.7. HER2

*HER2* is a gene located at the long arm of chromosome 17. It encodes an RTK protein of the ERBB family. HER2 has the highest activity among the receptors of this family, has no known ligands but can directly modulate *EGFR* signaling by combining with one of all the other members of the ERBB family to activate downstream MAPK, P3K/Akt, and JAK-STAT signaling pathways [111]. *HER2* alterations in NSCLC include gene mutations and amplifications. *HER2* mutations and *HER2* amplifications have been reported in 2–3% and 2–5% of lung adenocarcinomas, respectively [106,112]. Nevertheless, *HER2* was found to be amplified in 12–13% of NSCLC cases that have acquired resistance to *EGFR*-TKIs [113]. *HER2* driver mutations are invariably restricted to the first four exons (exons 18–21) of the tyrosine kinase domain of the gene; they have striking similarities to *EGFR* alterations concerning mutation type and specific clinico-pathological features [111,112,114,115]. The most commonly observed *HER2* mutation is an in-frame 12-bp insertion in exon 20 at the identical corresponding region of *EGFR*. It induces cell proliferation, motility, and survival processes [116,117].

*HER2* mutations are more prevalent in patients of Asian ethnicity than in other populations. A significant association with never-smoker status and female sex has been reported [114,115,117], while *HER2* amplifications seem to be more common in males and former-smokers [118]. In addition, *HER2* driver mutations and amplifications do not commonly overlap in the same patient, as no *HER2*-mutant tumors have been found to be amplified, which is a finding consistent with The Cancer Genome Atlas (TCGA) data [75]. This evidence suggests that *HER2* driver mutations actually represent distinct pathogenic entities and different molecular targets, and they should not be both referred to as “*HER2*-positive lung cancer” [115]. However, series of cases harboring both *HER2* amplification and mutation have also been sporadically reported [119]. The presence of an *HER2* mutation correlates with adenocarcinoma subtype, and the mutation rate seems to differ among lung adenocarcinoma subtypes, being mostly observed in acinar predominant adenocarcinoma (APA), papillary predominant adenocarcinoma (PPA), minimally invasive adenocarcinoma (MIA), and invasive mucinous adenocarcinoma (IMA) [112]. There are also some sporadic reports of *HER2* mutations in SCCs [115,117].

The impact of *HER2* dysregulation in the prognosis of NSCLC remains controversial, with inconsistent data from small-sized studies [112]. It has been suggested that the HER2 gene copy number may negatively affect sensitivity to anti-*EGFR* agents, being responsible for acquired resistance to *EGFR*-TKIs [111,118]. Further studies are necessary to better elucidate the prognostic roles of different genetic alterations of *HER2* in different subsets of naive and treated patients.

## 3. Co-Occurrence of Druggable Genetic Alterations in NSCLC

Co-occurring targetable driver mutations in *EGFR*-mutated NSCLS are a rare but well-documented event that negatively affects responses to TKIs and the subsequent clinical outcomes [39]. The presence of an *EGFR* mutation was thought, for a long time, to be mutually exclusive with driver alterations in the *KRAS* gene [32]. This is because the two genes share the same downstream signaling pathways, and a double activation should not hold any evolutionary benefit for lung cancers carrying double mutations. However, there is now evidence of coexisting *EGFR* and *KRAS* mutations in about 1.2% of NSCLCs, even though the single prevalence of the two genes is associated with quite divergent demographical subsets in terms of ethnicity, sex, and smoking status [28]. Similarly, also the coexistence of *EGFR* mutations and *ALK* rearrangements has been considered virtually impossible [120], up until concomitant alterations in both genes were documented in 1.3–1.6% of NSCLCs and were associated with poorer response to TKIs [39,121,122,123,124]. *ROS1* rearrangement usually occurs without other known oncogenic drivers, although there have been rare reported cases of concurrent mutations such as *EGFR*, *KRAS*, *BRAF*, *MET*, and *PIK3CA*; the clinical impact of such a co-occurring mutations is not yet clear.

In a large cohort of patients with *KRAS*-mutated NSCLC, mutations in *TP53* (39.4%), *STK11* (19.8%), *KEAP1* (12.9%), *ATM* (11.9%), *MET* (15.4%), and *HER2* amplifications (13.8%), only few rare co-occurring targetable mutations were found in *EGFR* (1.2%) and *BRAF* (1.2%) genes [85]. The presence of a co-occurring mutation was significantly associated with advanced tumor stage but showed no relation to smoking status, age, sex, or histologic subtype. Non-targetable *TP53* mutations are the most common co-occurring mutations, not-only in *KRAS*-mutated patients (approximately 50%) but also in those harboring *EGFR* mutations (22–40%), and have been associated with worst prognosis after treatment with TKIs [125,126]. Nevertheless, some variants of *TP53* mutations impact the oncological outcomes of patients treated with immunotherapy [127]. Therefore, the molecular characterization of *TP53* mutations may be necessary in the future, along with that of targetable genes, to better stratify patients in accordance with the types of co-occurring genetic alterations. Other, currently non-targetable mutations often occur in *EGFR*-mutated patients, with a negative prognostic impact, but their effect needs to be better clarified in future studies [126].

Co-occurring driver mutations in *BRAF*-positive NSCLC have been described with rates up to 13%, including *KRAS* (previously considered mutually exclusive), *EGFR* mutations, *ALK* fusions, *ROS1* fusions, *HER2* amplifications, and *MET* alterations. In most cases, concurrent mutations are found to be associated with class II and III *BRAF* mutations and are more rarely seen in NSCLC-harboring class I V600 mutations [98,99,101]. *TP53* is, again, the most frequent non-targetable mutation, co-occurring in about 30% of patients with tumors harboring *BRAF* mutations, but its clinical significance remains to be elucidated [128].

Traditional epidemiological studies present relevant limitations in investigating the patterns of co-occurrence of significant molecular alterations in NSCLC; in particular, some of the mutations under investigation are rare and often non-tested or found in limited cohorts, and others are missed because of a lack of large panels that are able to detect a wide range of mutations, amplifications, and other molecular aberrations. The advent of next-generation sequencing and other modern technologies will certainly allow for a better comprehension of the complex patterns of incidence of such aberrations, as well as their relations with several demographic, clinical, and pathological features of NSCLC. Furthermore, the pathogenic significance of concurrent molecular alterations, the impact of each single alteration, and the synergistic effects need to be further elucidated in future studies.

## 4. Conclusions

Different epidemiological patterns of targetable molecular alterations in NSCLC have been described worldwide. *EGFR* mutations have been detected in up to 50% of cases in Asian populations and only in 10–16% of Caucasians, and they mainly involve women without smoking history. *ALK* rearrangements seem to be more frequent in Asian and Pacific areas and are found commonly in younger never-smoking individuals. Similar features have also been reported for *ROS1* alterations, which, in addition, are more often found in advanced stage adenocarcinomas. *KRAS* mutations are the most frequent genetic alterations in NSCLC, as they occur in about 30% of cases; they are mutually exclusive with EGRF mutations, and, therefore, they are more common in Western populations and in long-term smokers. The presence of *KRAS* mutations, along with that of *BRAF* mutations, has been considered a negative prognostic factor, while MET genetic alterations have been associated with better prognoses and seem to occur in patients undergoing specific targeted therapies. The patterns of the co-occurrence of druggable genetic alterations in NSCLC are partially clarified and constantly updated as long as comprehensive data from studies performed with next-generation sequencing or other modern approaches are published. This is essential in order to fully elucidate the landscape of the molecular epidemiology and its clinical implications in NSCLC.

## Figures and Tables

**Figure 1 ijms-22-00612-f001:**
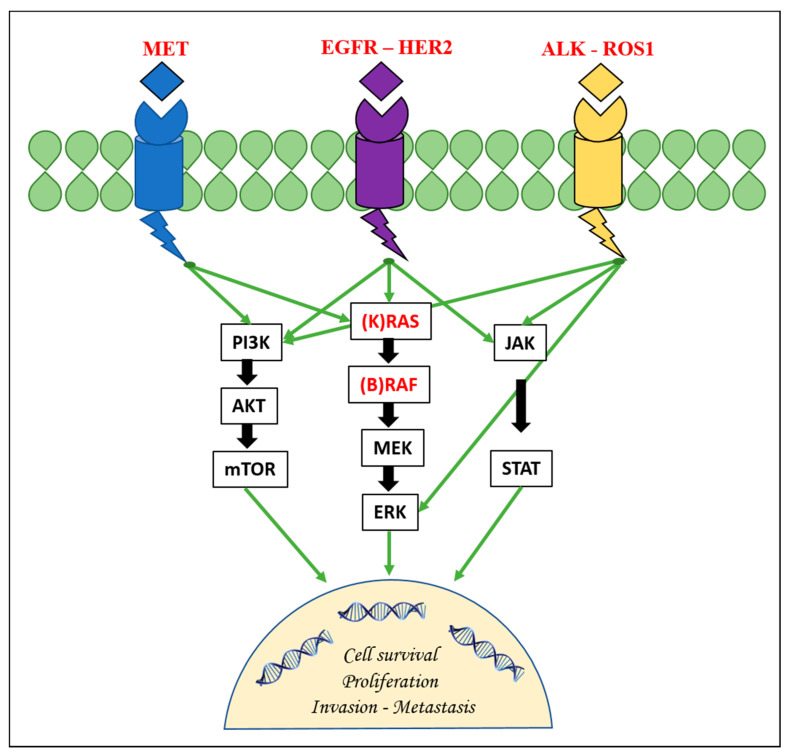
The figure depicts the main druggable genetic targets and their involvement in the main signaling pathways in non-small cell lung cancer. EGFR: epidermal growth factor receptor; HER2: human epidermal growth factor receptor 2; ERK: extracellular signal-related kinase.

**Table 1 ijms-22-00612-t001:** Main epidemiological features of molecular alterations in main candidate genes in non-small cell lung cancer.

Gene	Most Common Variants	Prevalence	Age	Gender	Smoking	Histology	Prognostic Significance
*EGFR*	Mutations in exons 19 and 21	10–16% in Western populations, 40–50% in Asians.	Younger patients	Females	Never smokers	Adenocarcinoma	Response to specific TKIs, T790M predictor of resistance
*ALK*	EML4-*ALK* variants	1–10% of NSCLC	Younger patients	Females? Not clear	Never smokers	Adenocarcinoma	Aggressive tumors, response to specific TKIs
*ROS1*	CD74-*ROS1* variants	0.9–2.6% of NSCLC	Younger patients	Females	Never smokers	Adenocarcinoma	Less aggressive tumors, response to specific TKIs
*KRAS*	Mutations in codons 12 and 13	30–40% of NSCLC, more common in Caucasians	Older ages	Males? Not clear	Smokers	Adenocarcinoma	Poor prognosis or response to TKIs? Not clear
*BRAF*	Mutations in exon 15	2–4% of NSCLC	No predilection	V600E in females and others in males	Smokers	Adenocarcinoma	Aggressive tumors and poor prognosis? Not clear. Response to *BRAF* inhibitors.
*MET*	Mutations in exon 14, amplification	Mutations in 1–10% of NSCLC, amplification in 5–22%	Older ages	Not clear	Smokers	Adenocarcinoma and SCC	Resistance to *EGFR*-TKIs. Response to *MET* inhibitors.
*HER2*	Mutations in exons 18–21, amplification	Mutations in 2–3% and amplifications in 2–5% of adenocarcinomas	Not clear	Mutations in females and amplifications in males	Mutations in never smokers and amplifications in ex-smokers	Adenocarcinoma Rare in SCC	Resistance to *EGFR*-TKIs? Not clear

NSCLC: non-small cell lung cancer; SCC: squamous cell carcinoma; TKIs: tyrosine kinase inhibitors; EML4: echinoderm microtubule-associated protein like 4.

## Data Availability

Not applicable.

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
