# Peer review of "Molecular Epidemiology of the Main Druggable Genetic Alterations in Non-Small Cell Lung Cancer"

_ijms, 2021, doi:10.3390/ijms22020612_

Round 1
Reviewer 1 Report
Authors have put tremendous efforts in writing and compiling the data for the review article titled "Molecular epidemiology of the main druggable genetic alterations in non-small cell lung cancer."
Although the Authors have made enough efforts to make this article interesting, this article still needs some clarification before it is ready for publications; those are as follows:
Line no. 30-33: It would be good if the authors can provide more recent statistics about the new cases.
Line no. 32-36: Are there any genetic, geographical, or environmental factors in the countries the authors have mentioned for higher incidences?
Line no-37-43: What is the meaning of historically here? Is there any rationale for being these gender-specific cases?
Headings should be in sequence. Heading 2 EGFR does not fit just after the Introduction.
Table 1 is not very well explained; EGFR, KRAS has shown the geographical percentage while others did not; why?
Figure quality is not very clear.
The authors explained all about the background of genes rather than discussing their prominent role in Lung cancer. It would be better if authors can explain the epidemiology of NSCLC in detail.
The authors have not put extra emphasis on Druggable genetic alteration in any part of the article.
Make another table with the information of these genes specific to gender and smoking habits. If possible, explain why these variations are gender and ethnicity biased.
What are the chances of co-expression of these genes in lung cancer?
Which gene dominates the other one at what percentage. Are they expressed similarly or not? If not, what makes the difference?
Is there any case where most of these genes express at the same intensity?
Instead of putting a different heading of druggable genetic alterations, authors can add the same under each gene with a subheading.
There is less correlation between Introduction and overall article, and It would be good if authors can add more information on available drugs
Author Response
Dear reviewer
Enclosed please find the revised version of the manuscript by Sara Solveig Fois et al., entitled Molecular epidemiology of the main druggable genetic alterations in non-small cell lung cancer. We would like to thank you for your valuable suggestions. Our responses to your comments are listed here issue by issue. Changes in the manuscript are highlighted in yellow.
Issue.
Line no. 30-33: It would be good if the authors can provide more recent statistics about the new cases. The authors explained all about the background of genes rather than discussing their prominent role in Lung cancer. It would be better if authors can explain the epidemiology of NSCLC in detail.
Reply
Data about new cases were updated as of 2020, and the discussion on epidemiology has been expanded.
Issue.
Line no. 32-36: Are there any genetic, geographical, or environmental factors in the countries the authors have mentioned for higher incidences?
Reply
The most incident factor impacting the geographical distribution of cases is cigarette smoking. A comment was added in the text.
Issue.
Line no-37-43: What is the meaning of historically here? Is there any rationale for being these gender-specific cases?
Reply
The concept was rephrased to be clearer.
Issue.
Headings should be in sequence. Heading 2 EGFR does not fit just after the Introduction. Instead of putting a different heading of druggable genetic alterations, authors can add the same under each gene with a subheading.
Reply
The headings of the paper have been restructured as suggested.
Issue.
Table 1 is not very well explained; EGFR, KRAS has shown the geographical percentage while others did not; why?
Reply
Table 1 has been moved and better explained in the text. Geographical or other data are present when available.
Issue.
Figure quality is not very clear.
Reply
We added a new larger figure with higher quality for your assessment. In any case, upon acceptance of the paper, we will be able to improve the quality of the figure in collaboration with the production team of the journal.
Issue.
The authors have not put extra emphasis on Druggable genetic alteration in any part of the article. There is less correlation between Introduction and the overall article, and It would be good if the authors can add more information on available drugs.
Reply
The currently available and recommended drugs against the genes discussed have been added in a new paragraph in the “introduction”.
Issue.
Make another table with the information of these genes specific to gender and smoking habits. If possible, explain why these variations are gender and ethnicity biased.
Reply
Data regarding gender and smoking habits are discussed in the text and summarized in table one. It seems to us that making a further table only for sex and smoking it would probably be excessive and redundant.
Issue.
What are the chances of co-expression of these genes in lung cancer? Which gene dominates the other one at what percentage. Are they expressed similarly or not? If not, what makes the difference? Is there any case where most of these genes express at the same intensity?
Reply
Unfortunately, not all these questions can be answered on the basis of the current evidence. We added some further comments on gene co-expression and their interaction with TP53 and other common non-targetable genes in the text. In most cases, the current evidence is not enough to explain the clinical impact of such co-occurrence.
Hoping to have addressed all the issues, we are looking forward to receiving news from You.
Yours Sincerely,
Dr. Panagiotis Paliogiannis
Reviewer 2 Report
- Fois and coworkers describe in detail the commonly found druggable mutations in NSCLC. The authors are very detailed and thorough in their discussion and focus on druggable mutations from a clinical significance perspective. The authors describe in detail common mutations such as EGFR, KRAS, BRAF, MET, HER2 mutations or amplification, ALK and ROS1 fusion.
- While the authors talk about druggable mutations throughout the manuscript, the authors never mention any information related to drugs or therapeutics. The authors should add value to the paper by including atleast one paragraph related to commonly used drugs that target these mutations. The authors should also cite relevant papers in the field:
- Arbor et al., JAMA. 2019;322(8):764-774. doi:10.1001/jama.2019.11058
- Cheng et al., J Thorac Dis. 2020 Mar; 12(3): 1056–1069
- Singh et al., Bioorganic & Medicinal Chemistry 27 (2019) 3477–3510
- Herbst et al., Nature volume 553, pages446–454(2018)
- Singh et al., Anti-Cancer Agents in Medicinal Chemistry (Formerly Current Medicinal Chemistry - Anti-Cancer Agents), Volume 19, Number 7, 2019, pp. 842-874(33)
- Minor changes in sentence structure and grammar are required throughout the manuscript. Eg: druggable spelled as “draggable” in several places, and other such similar errors
Author Response
Dear reviewer
Enclosed please find the revised version of the manuscript by Sara Solveig Fois et al., entitled Molecular epidemiology of the main druggable genetic alterations in non-small cell lung cancer. We would like to thank you for your valuable suggestions. Our responses to your comments are listed here issue by issue. Changes in the manuscript are highlighted in yellow.
Issue
While the authors talk about druggable mutations throughout the manuscript, the authors never mention any information related to drugs or therapeutics. The authors should add value to the paper by including atleast one paragraph related to commonly used drugs that target these mutations.
Reply
The currently available and recommended drugs against the genes discussed have been added in a paragraph in the introduction section.
Issue
The authors should also cite relevant papers in the field:
Arbor et al., JAMA. 2019;322(8):764-774. doi:10.1001/jama.2019.11058
Cheng et al., J Thorac Dis. 2020 Mar; 12(3): 1056–1069
Singh et al., Bioorganic & Medicinal Chemistry 27 (2019) 3477–3510
Herbst et al., Nature volume 553, pages446–454(2018)
Singh et al., Anti-Cancer Agents in Medicinal Chemistry (Formerly Current Medicinal Chemistry - Anti-Cancer Agents), Volume 19, Number 7, 2019, pp. 842-874(33)
Reply
The papers suggested have been added in the manuscript.
Issue
Minor changes in sentence structure and grammar are required throughout the manuscript. Eg: druggable spelled as “draggable” in several places, and other such similar errors.
Reply
Minor spelling and typing errors have been corrected throughout the text.
Hoping to have addressed all the issues, we are looking forward to receiving news from You.
Yours Sincerely,
Dr. Panagiotis Paliogiannis
Round 2
Reviewer 1 Report
The authors have answered most of the questions raised by reviewers. Few of the reviewer's concerns have not dealt with the article's present form, which the authors have justified in response. There no further revision needed.